# Recurrence Pattern Is an Independent Surgical Prognostic Factor for Long-Term Oncological Outcomes in Patients with Hepatocellular Carcinoma

**DOI:** 10.3390/biomedicines12030655

**Published:** 2024-03-14

**Authors:** Heng-Yuan Hsu, Jui-Hsiang Tang, Song-Fong Huang, Chun-Wei Huang, Sey-En Lin, Shu-Wei Huang, Chao-Wei Lee, Tsung-Han Wu, Ming-Chin Yu

**Affiliations:** 1Department of Surgery, New Taipei Municipal TuCheng Hospital (Built and Operated by Chang Gung Medical Foundation), New Taipei City 23652, Taiwan; smallredshoe@gmail.com (H.-Y.H.); b9102059@cgmh.org.tw (S.-F.H.); n740227@cgmh.org.tw (C.-W.H.); 2Division of Gastroenterology and Hepatology, Department of Internal Medicine, New Taipei Municipal TuCheng Hospital (Built and Operated by Chang Gung Medical Foundation), New Taipei City 23652, Taiwan; gima2239@cgmh.org.tw (J.-H.T.); huangshuwei@gmail.com (S.-W.H.); 3Department of Pathology, New Taipei Municipal TuCheng Hospital (Built and Operated by Chang Gung Medical Foundation), New Taipei City 23652, Taiwan; linse@cgmh.org.tw; 4Department of Surgery, Linkou Chang Gung Memorial Hospital, Taoyuan City 33305, Taiwan; alanchaoweilee@hotmail.com (C.-W.L.); wutsunghan@gmail.com (T.-H.W.); 5Institute of Clinical Medical Sciences, College of Medicine, Chang Gung University, Taoyuan City 33305, Taiwan

**Keywords:** hepatocellular carcinoma, hepatectomy, recurrence pattern, oncological outcome

## Abstract

Background: The perioperative outcomes of a partial hepatectomy for hepatocellular carcinoma (HCC) have improved. However, high recurrence rates after a curative hepatectomy for HCC is still an issue. This study aimed to analyze the difference between various recurrence patterns. Methods: We retrospectively reviewed 754 patients with HCC who underwent a curative hepatectomy between January 2012 and March 2021. Patients with recurrent events were categorized into three types: regional recurrence (type I), multiple intrahepatic recurrence (type II), or presence of any distant metastasis (type III). Results: The median follow-up period was 51.2 months. Regarding recurrence, 375 (49.7%) patients developed recurrence, with 244 (32.4%), 51 (6.8%), and 80 (10.6%) patients having type I, II, and III recurrence, respectively. Type III recurrence appeared to be more common in male patients and those with major liver resection, vascular invasion, a large tumor size (>5 cm), a higher tumor grade, and higher levels of AST and AFP (*p* < 0.05). Patients who had distant metastasis at recurrence had the shortest recurrence time and the worst overall survival (*p* < 0.001 and *p* < 0.001). Conclusions: our study demonstrated that recurrence with distant metastasis occurred earliest and had the worst outcome compared to regional or multiple intrahepatic recurrences.

## 1. Introduction

Hepatocellular carcinoma (HCC) is the fifth most common malignant disease and the second leading cause of cancer deaths worldwide [1]. Although the perioperative mortality and short-term outcomes of HCC after a curative hepatectomy have improved, HCC recurrence is still a challenge [2,3]. There is no definite guideline for the management of recurrent HCC. However, either repeated hepatectomy or salvage liver transplantation are good surgical strategies for the management of selected patients with intrahepatic recurrence. Moreover, systemic immunotherapies combined with target therapies have also shown promising outcomes in a comparative open-label randomized study [4]. The multi-omics study demonstrated the different patterns between the multicentric origin and intrahepatic metastases. It also proved that tumor aggressiveness and biology were important factors that influenced treatment outcomes [5]. Therefore, treatment should be based on the location, size, and number of lesions in cases of cancer recurrence. Systemic therapies should also be considered in patients with an advanced-stage disease or those beyond the up-to-seven criteria [6]. In the real world, because of the limitations of ablation in terms of tumor size, tumor number, and sink effect, surgical resection still remains the main first-line treatment for HCC. Regarding liver transplantation, the scarcity of grafts is the main challenge.

Numerous studies have focused on pathologic, radiologic, and biologic data, and many predicting systems have been developed [7,8,9]. The treatment design should be delicately discussed by a multidisciplinary team to incorporate ablation, trans-arterial chemoembolization (TACE), and surgery. However, the pattern of recurrence is still hard to predict using clinical parameters. The appropriate surgical margin is also another topic of debate. In most of the literature, recurrence within 2 years after a curative hepatectomy is defined as early recurrence [10,11]. Most of the reviewed literature focuses on the width of the safety margin to prevent early recurrence. However, we previously demonstrated that narrow surgical margins have comparable outcomes to those of wider surgical margins [12]. Anatomical hepatectomies were more prevalent in eastern countries; nevertheless, their adoption remains controversial due to surgeon-related confounding factors and challenges associated with establishing a precise definition. Moreover, patients’ outcomes could be better in cases with smaller tumors [13,14]. Although minimal invasive surgery offers superior short-term outcomes when compared with open surgery, its oncologic outcomes need to be carefully examined [15]. However, the clinical association between the recurrence pattern and the anatomic and pathologic variables are rarely studied. Furthermore, the relationship between recurrence and long-term outcomes was need to be discussed.

In the current study, we investigate and analyze different types of recurrence in patients with HCC after a curative hepatectomy to demonstrate the importance of tumor biology.

## 2. Materials and Methods

### 2.1. Study Population

We retrospectively review patients with HCC who were treated with curative hepatectomies by the same surgical team at Chang Gung Memorial Hospital (CGMH) and TuCheng Municipal Hospital (built and operated by Chang Gung Medical Foundation). A total of 764 patients were reviewed between January 2012 and March 2021. This study was approved by the Institutional Review Boards (CGMH IRB No: 201801550B0) of Chang Gung Memorial Hospital (CGMH). All surgical specimens were sent to and checked by qualified pathologists for final diagnosis. Tumor staging was based on the 8th edition of the AJCC TNM staging system for HCC [16]. The exclusion criteria were cases of combined hepatocellular-cholangiocarcinoma, recurrent HCC, non-curative intent hepatectomies, the presence of distant metastases, preoperative anticancer treatment, or patients with a history of other malignant diseases. Regarding in-hospital mortality, 10 (1.3%) patients who developed perioperative mortality 30 days postoperatively were also excluded from our study cohort. The operative methods included laparotomy, minimally invasive surgery, or hybrid based on the IWATE criteria and the surgeon’s preference. Overall, 754 patients were enrolled in the current study. The patients’ clinical and pathological variables are summarized in Table 1.

### 2.2. Definition of Recurrence and Recurrent Pattern

All patients underwent blood test and triphasic computed tomography (CT) of the liver within 1 month after liver resection. Regular cancer follow-up including liver imaging, liver function test, and serum α-fetoprotein (AFP) levels were surveyed every 3 months in the first 2 years and every 4–6 months thereafter. When recurrent event was suspected, serially delicate studies and restaging were conducted. Definite recurrence was defined by the presence of typically dynamic imaging findings, elevated serum AFP levels, or biopsy. Treatment was started based on the suggestions of a multi-modality liver cancer team. Patients were followed up until 31 March 2023.

All cases of recurrence were reviewed and discussed by our cancer team. In the current study, we divided all recurrent events into three types: Type I recurrence was defined as a solitary intrahepatic lesion, which could be considered re-resection technically. Type II recurrence was defined as multiple intrahepatic metastases, while type III recurrence referred to cases that presented with any distant metastasis such as lung, bone, or lymph node metastasis.

### 2.3. Statistical Analysis

Student’s *t* test was used to compare continuous variables, while Pearson’s χ2 test was applied to analyze categorical variables. Survival curves of different types of recurrence and overall survival were drawn using the Kaplan–Meier method and compared using log-rank test. Risk factor for recurrence was surveyed using Cox regression analysis. All significant factors in the univariate analysis were included in the multivariate analysis. Results from the multivariate analysis were demonstrated as hazard ratios (HR) and 95% confidence intervals (CI). Statistical analysis was performed using SPSS statistics version 21.0 (IBM Corp., Somers, NY, USA). Two-tailed *p*-values < 0.05 were considered statistically significant.

## 3. Results

### 3.1. Patients’ Clinical and Pathological Variables

Overall, 754 patients were included in this study. The patients’ demographic data were as follows: 76.7% were male, 26.5% had diabetes, 59.3% had HBV, 26.7% had HCV, and most were Child A patients. The mean age of the study cohort was 61.9 ± 11.0 years. A total of 375 (49.7%) patients had recurrence, while 379 (50.3%) patients were disease-free. The median follow-up duration was 51.2 ± 31.7 months and 31.5% of patients had cancer-specific or liver-related mortality during the long-term follow-up (Figure 1). The mean tumor size was 4.5 ± 3.2 cm, while in 212 (28.1%) patients, the tumor size was more than 5 cm (Table 1). The overall survival rates for patients who were disease-free and those with recurrence were 96.0% and 58.2% at 5 years and 89.5% and 35.2% at 10 years, respectively (*p* < 0.05). Overall, 104 (13.8%), 299 (39.7%), 227 (30.1%), 42 (5.6%), and 82 (10.9%) patients were defined as stage Ia, Ib, II, IIIa, and IIIb, respectively. The 5 years DFS rate was 64.0, 51.2, 38.6, 16.8, and 29.0%, respectively (Figure 1). The odd ratios analysis focused on host, surgical, and pathological factors which showed male gender, cirrhosis, vascular invasion, and satellite lesions were the independent factors for HCC recurrence after resection (Table 2).

### 3.2. Comparison of Recurrence Patterns

Regarding recurrence, 244 patients (32.4%) who had solitary intrahepatic lesions that could be considered re-resection technically were classified as having regional recurrence (Type I). There were 51 (6.8%) and 80 (10.6%) patients with type II and type III recurrence, which implied the presence of multiple intrahepatic lesions and distant metastasis, respectively (Table 1). Type III recurrence appeared to be more common in male patients and those who had major liver resection, vascular invasion, a large tumor size (>5 cm), a higher tumor grade, and higher levels of AST and AFP (Table 1, *p* < 0.05). Furthermore, 139 (57.0%), 23 (45.1%), and 40 (50.0%) patients had cirrhotic livers in type I, II, and III recurrence groups, respectively (*p* = 0.001). Moreover, type III recurrence was associated with AJCC stage III (*p* < 0.001). The recurrence time after resection was 24.8 ± 23.0, 15.0 ± 14.1, and 10.9 ± 12.8 months for type I, II, and III recurrence, respectively (Figure 2
*p* < 0.001 by ANOVA). The overall survival time was 110.30 ± 3.14, 34.52 ± 7.10, and 30.02 ± 8.35 months for type I, II, and III recurrence, respectively (*p* < 0.001).

### 3.3. Analysis of Disease-Free Survival and Overall Survival

Of the 754 patients, 375 (49.73%) had recurrent events and had worse overall survival outcomes compared to those without recurrence (Figure 2). Patients without cancer recurrence had better long-term outcomes compared to those who had recurrence (Figure 1, *p* < 0.001). After COX regression multivariate analysis, the independent prognostic factors for DFS were male sex, diabetes, higher ICG-R15, large tumor size, vascular invasion, and cirrhosis (Table 3). The recurrence patterns analysis for OS analysis demonstrated that diabetes, the presence of grade III or more surgical complications, higher initial tumor staging, and an aggressive recurrence pattern were significant risk factors for a poor oncological outcome (Table 4). In summary, aggressive tumor biological factors were associated with early recurrences and the presence of distant metastases.

### 3.4. Higher Tumor Grade Was Associated with Higher Distant Metastasis Risks

The tumor grade was analyzed with the recurrence pattern and we completed statistical analyses with two tiers and four tiers (G1 + G2 vs. G3 + G4 and G1–G4). The results are summarized in Table 5. The higher tumor grade had a higher distant metastasis risk. Most of the literature has mentioned that the tumor grade was dichotomized as a lower and higher grade and was associated with a poor outcome [17]. The higher tumor grade also had higher tumor genome instability. The copy number aberration analysis showed an increased chromosome imbalance from the normal tissue. The higher chromosome imbalance was an independent factor in the HCC outcome [18]. The impact of tumor biology characters included the size, number, satellite lesion, rupture, vascular invasion, and tumor grade; however, the tumor grade is not listed in the AJCC staging system.

## 4. Discussion

Recurrence is consistently a crucial concern in curative treatment for HCC, prompting studies aimed at predicting the likelihood of recurrence. The ERASL model that included over 3000 cases showed that the male sex, ALBI grade, microscopic invasion, AFP, tumor size, and number of lesions could be incorporated to predict the possibility at a cut-off value; furthermore, the model indicated that risk factors could be stratified from low- to high-risk groups [19,20,21]. Higher serologic factors, such as AFP and des-γ-carboxyprothrombin (DCP), tumor volume, multiple nodules, microvascular invasion, and poorly differentiated types have been assumed to be aggressive tumor factors. Moreover, a multicenter study validated the value of the ADV score in Korea and Japan [22,23]. Although anatomical hepatectomy has been proposed to improve the clinical outcome of HCC, it is more popular in eastern countries [24]. A positive resection margin should be prevented as much as possible because of higher recurrence rates; however, there is no clear difference in the long-term outcomes between wide and narrow margins [12,25]. Postoperative higher-grade complications and blood transfusions related to poor surgical events should also be prevented [9,26]. However, only a few papers have analyzed the impact of different recurrence patterns on long-term outcomes.

A salvage hepatectomy was assumed to be a standard treatment option for local recurrence after radiofrequency ablation; nevertheless, it was shown to have a minimal benefit in cases of recurrence associated with aggressive tumors [27,28]. Furthermore, although repeated surgical resection is an important strategy to improve patients’ outcomes, the rate of extrahepatic metastases has been reported to be 2.4–18.0% in some literature reviews [29,30,31,32]. Extrahepatic recurrence occurs more commonly in cases of intermediate and advanced HCC, and the lungs and bones are common sites of metastasis [33]. Our study’s findings show that type III recurrence usually occurs within the first year after resection. To improve the treatment outcome, an aggressive hepatectomy should be considered for early-stage HCC with good tumor biology, while systemic treatment should be considered for cases of advanced HCC with worse tumor biology.

This study demonstrated that distant metastases could be encountered after curative hepatectomies and tended to occur earlier than other recurrence patterns. Moreover, 10.6% of patients with type III recurrence and 6.8% of those with type II recurrence tended to develop a systemic disease earlier and had poor survival outcomes. Therefore, neoadjuvant or adjuvant therapy should be considered in patients with these aggressive tumor statuses. The administration of sorafenib in a phase III adjuvant therapy trial did not lead to a significant difference in recurrence-free survival [34]. However, a meta-analysis of data from 2655 patients who underwent adjuvant therapy revealed that the administration of sorafenib effectively prolonged overall survival and reduced the recurrence rate [35]. Following the favorable outcomes observed with atezolizumab plus bevacizumab in the adjuvant context, as reported at the 2023 AACR conference, the overall understanding of the molecular pathogenesis of HCC has advanced, revealing alterations in tumor drivers leading to genomic instability. This knowledge of molecular and immune subclasses could serve as a means to assess tumor aggressiveness [36]. The neutrophil-to-lymphocyte ratio and platelet-to-lymphocyte ratio have been reported as independent prognostic factors in cases of combined treatment; however, immune escape and resistance to anti-PD-1 therapy through β-catenin activation have also been reported [37,38].

Our study has some limitation including its retrospective design and small patient numbers. Selection bias was also inevitable and could have affected the statistical results. All the patients in our study were disease-free after a curative hepatectomy. The role of adjuvant therapy could not be determined in this cohort, as it had not been initiated.

## 5. Conclusions

In conclusion, our study demonstrated that recurrence with distant metastasis after a partial hepatectomy was a poor independent prognostic factor; moreover, it appears to occur earlier and has a worse outcome compared to other patterns of recurrence. Neoadjuvant or adjuvant systemic therapies could be considered in selected patients. More prospective and well-designed trials are also needed to improve patients’ outcomes.

## Figures and Tables

**Figure 1 biomedicines-12-00655-f001:**
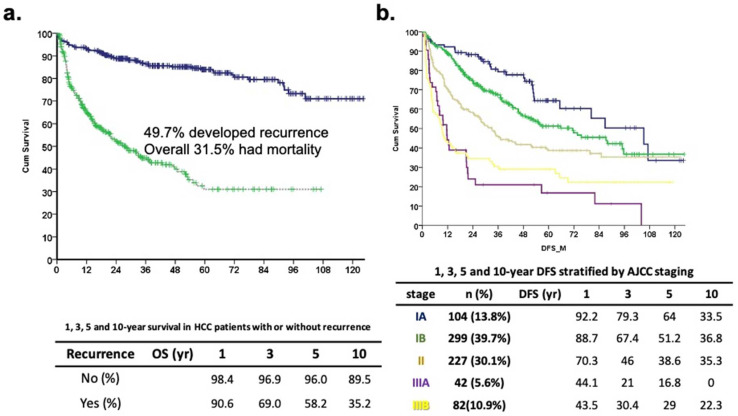
Long-term outcome of study cohort. (**a**) Disease-specific survival curve (blue line) and disease-free survival curve (green line) of 754 HCC patients. The 1-, 3-, 5-, and 10-year disease-specific survival was 98.4, 96.9, 96.0, and 89.5 months in no recurrence group and 90.6, 69.0, 58.2, and 35.2 months in recurrence patients, respectively (*p* < 0.001). (**b**) Disease-free survival curve stratified by AJCC staging (*p* < 0.001).

**Figure 2 biomedicines-12-00655-f002:**
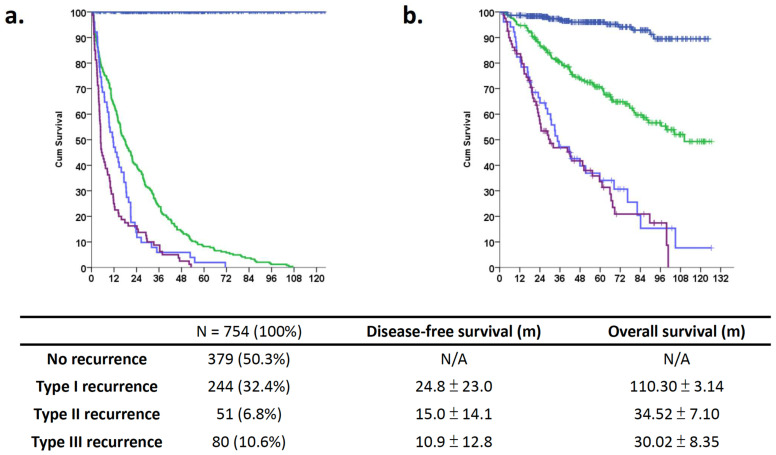
Long-term outcomes of different recurrence patterns among patients with hepatocellular carcinoma who underwent curative hepatectomy. Different long-term outcomes of patients with HCC stratified according to different recurrence patterns: blue line for no recurrence, green line for intrahepatic single, light blue for multiple intrahepatic, and purple for multiple distant metastasis. (**a**) The life table shows the recurrence-free survival curve for patients without recurrence and those with different types of recurrence. The recurrence time after resection was 24.8 ± 23.0, 15.0 ± 14.1, and 10.9 ± 12.8 months for type I, II, and III recurrences, respectively (*p* < 0.001 according to ANOVA). (**b**) Kaplan–Meier analysis shows that the 5-year overall survival differ among the recurrence groups. The overall survival time was 110.30 ± 3.14, 34.52 ± 7.10, and 30.02 ± 8.35 months for type I, II, and III recurrence patterns, respectively (*p* < 0.001).

**Table 1 biomedicines-12-00655-t001:** Demographic data of 754 HCC patients, along with a comparison of recurrence patterns.

Variables	All (*n* = 754)	No (*n* = 379)	TYPE I (*n* = 244)	TYPE II (*n* = 51)	TYPE III (*n* = 80)	*p* Value
	100%	49.7%	32.4%	6.8%	10.6%	
Age (years)	61.9 ± 11.0	62.1 ± 11.1	62.1 ± 10.7	60.3 ± 11.1	61.7 ± 11.9	0.730
Gender (male)	578 (76.7)	274 (72.3)	198 (81.1)	40 (78.4)	66 (82.5)	0.039 *
Comorbidity (yes)	444 (58.4)	210 (55.4)	155 (63.56)	33 (64.7)	42 (52.5)	0.111
HBV-positive	447 (59.3)	218 (57.5)	148 (60.7)	33 (64.7)	48 (60.0)	0.725
HCV-positive	201 (26.7)	100 (26.4)	73 (29.9)	9 (17.6)	19 (23.8)	0.283
ICG R15	10.2 ± 8.0	9.7 ± 7.5	11.6 ± 10.4	9.6 ± 7.1	10.8 ± 9.7	0.073
Major hepatectomy	228 (30.2)	94 (24.8)	67 (27.5)	24 (47.1)	43 (53.8)	<0.001 ***
Anatomic resection	393(52.3)	198 (52.4)	122 (50.0)	22 (43.1)	51 (65.4)	0.055
Complication (yes)	50 (6.7)	22 (5.8)	16 (6.6)	5 (9.8)	7 (8.8)	0.618
AST (IU/L)	42.8 ± 26.7	39.5 ± 23.2	44.4 ± 27.1	49.2 ± 29.9	49.6 ± 35.7	0.002 **
ALT (IU/L)	42.9 ± 36.8	40.8 ± 34.1	45.2 ± 41.5	46.7 ± 31.6	43.4 ± 36.9	0.428
BIL (mg/dL)	0.7 ± 0.3	0.7 ± 0.3	0.7 ± 0.3	0.6 ± 0.2	0.8 ± 0.4	0.027 *
ALB (g/dL)	4.2 ± 0.4	4.2 ± 0.4	4.2 ± 0.4	4.1 ± 0.4	4.2 ± 0.4	0.069
AFP (ng/mL)	3329.0 ± 25,085.4	1468.7 ± 12,505.9	1969.7 ± 8287.0	4848.3 ± 20,970.3	15,319.6 ± 67,755.4	<0.001 ***
AFP (>1000 ng/mL)	82 (10.9)	30 (7.9)	31 (12.7)	6 (11.8)	18 (18.8)	0.195
Cirrhosis	355 (47.1)	153 (40.4)	139 (57.0)	23 (45.1)	40 (50.0)	0.001 **
Satellite lesion						<0.001 ***
No	668 (88.6)	357 (94.2)	208 (85.2)	39 (76.5)	64 (80.0)	
Single	53 (7.0)	14 (3.7)	19 (7.8)	10 (19.6)	10 (12.5)	
Multiple	33 (4.4)	8 (2.1)	17 (7.0)	2 (3.9)	6 (7.5)	
Vascular invasion						<0.001 ***
No	481 (63.8)	279 (73.6)	149 (61.1)	24 (47.1)	29 (36.3)	
Microscopic	233 (30.9)	91 (24.0)	78 (32.0)	24 (47.1)	40 (50.0)	
Gross	40 (5.3)	9 (2.4)	17 (7.0)	3 (5.9)	11 (13.8)	
Margin < 0.5 cm	408 (54.1)	195 (51.5)	143 (58.6)	28 (54.9)	42 (52.5)	0.367
Tumor size > 5 cm	212 (28.1)	79 (20.8)	59 (24.2)	26 (51.0)	48 (60.0)	<0.001 ***
Tumor size (cm)	4.5 ± 3.2	3.9 ± 2.8	4.4 ± 3.1	5.9 ± 3.6	6.9 ± 3.9	<0.001 ***
Rupture	48 (6.4)	15 (4.0)	12 (4.9)	8 (15.7)	13 (16.3)	0.074
Grade III, IV	308 (41.0)	138 (36.4)	104 (43.2)	19 (37.3)	47 (58.8)	0.002 **
AJCC 8 staging I II III	403 (53.5) 227 (30.1) 124 (16.4)	249 (65.7) 100 (26.4) 30 (7.9)	115 (47.1) 85 (34.8) 44 (18.0)	18 (35.3) 15 (29.4) 18 (35.3)	21 (26.3) 27 (33.8) 32 (40.0)	<0.001 ***

* statistical significance (* *p* < 0.05, ** *p* < 0.01, *** *p* < 0.001) HBV: hepatitis B virus; HCV: hepatitis C virus; AST: aspartate aminotransferase; ALT: alanine aminotransferase; BIL: bilirubin; ALB: albumin; AFP: alpha-fetoprotein; AJCC 8 staging: the 8th edition of American Joint Committee on Cancer TNM staging system.

**Table 2 biomedicines-12-00655-t002:** Risk factor analysis between recurrence and non-recurrence patients after curative hepatectomy.

Variables	Univariate Analysis		Multivariate Analysis		
	Odds Ratio	*p* Value	Odds Ratio	95% CI	*p* Value
Age (60 years) (>60 vs. ≤60)	1.071	0.643			
Sex (M/F) (M vs. F)	1.642	0.004 **	1.767	1.220–2.558	0.003 **
Comorbidity (Yes vs. No)	1.277	0.099			
AST (68 U/L) (Higher vs. lower)	1.939	0.004 **	1.468	0.896–2.405	0.127
ALT (72 U/L) (Higher vs. lower)	1.530	0.070			
AFP (1000 ng/mL) (Higher vs. lower)	1.873	0.009 **	1.274	0.742–2.188	0.380
Major hepatectomy (Yes vs. No)	1.686	0.001 **	1.182	0.802–1.739	0.399
Anatomic resection (Yes vs. No)	1.004	0.978			
Close margin (Yes vs. No)	1.241	0.141			
Complication (Yes vs. No)	1.453	0.726			
Tumor size (cm) (>5.0 vs. ≤5.0)	2.087	<0.001 ***	1.507	0.988–2.296	0.057
Satellite lesions (Yes vs. no)	3.339	<0.001 ***	1.930	1.096–3.398	0.023 *
Vascular invasion (Yes vs. no)	2.389	<0.001 ***	1.727	1.217–2.449	0.002 **
Grading I/ II/ III, IV (III, IV vs. I, II)	1.470	0.010 *	1.125	0.812–1.559	0.479
Tumor rupture (Yes vs. No)	2.342	<0.001 ***	1.590	0.780–3.239	0.202
Cirrhosis (Yes vs. No)	1.725	<0.001 ***	2.133	1.545–2.943	<0.001 ***

* statistical significance (* *p* < 0.05, ** *p* < 0.01, *** *p* < 0.001); 95% CI, 95% confidence interval of odds ratio; AST: aspartate aminotransferase; ALT: alanine aminotransferase; AFP: alpha-fetoprotein.

**Table 3 biomedicines-12-00655-t003:** Clinicopathologic Factors and Disease-Free Survival in 754 Patients with HCC.

Variables	Univariate Analysis			Multivariate Analysis		
	HR	95% CI	*p* Value	HR	95% CI	*p* Value
Age (60 years) >60 vs. ≤60	1.136	0.924–1.395	0.225			
Sex (M/F) M vs. F	1.435	1.109–1.859	0.006 **	1.634	1.247–2.151	<0.001 ***
Diabetes Yes vs. No	1.356	1.084–1.696	0.008 **	1.300	1.033–1.637	0.026 *
Comorbidity Yes vs. No	1.209	0.982–1.489	0.074			
ICG-R15 (10%) Higher vs. lower	1.364	1.066–1.746	0.014 *	1.380	1.055–1.804	0.019 *
AST (68 U/L) Higher vs. lower	1.678	1.265–2.226	<0.001 ***	1.217	0.806–1.835	0.350
ALT (72 U/L) Higher vs. lower	1.380	1.022–1.864	0.036 *	1.031	0.678–1.568	0.887
α-fetal protein; AFP (1000 ng/mL) Higher vs. lower	1.654	1.233–2.218	0.001 *	1.073	0.763–1.508	0.687
Major hepatectomy Yes vs. No	1.631	1.342–2.049	<0.001 ***	1.224	0.945–1.544	0.125
Anatomic resection Yes vs. No	1.092	0.891–1.339	0.397			
Close margin Yes vs. No	1.194	0.973–1.464	0.089			
Complication Yes vs. No	1.515	1.030–2.238	0.035 *	1.162	0.772–1.750	0.471
Tumor size (cm) >5.0 vs. ≤5.0	2.098	1.696–2.595	<0.001 ***	1.762	1.346–2.306	<0.001 ***
Satellite lesions Multiple vs. single vs. no	2.271	1.733–2.975	<0.001 ***	1.220	0.985–1.510	0.068
Vascular invasion Thrombus vs. microscopic vs. no	1.712	1.462–2.005	<0.001 ***	1.235	1.022–1.492	0.029 *
Grading I/ II/ III, IV III, IV vs. I, II	1.338	1.090–1.641	0.005 **	1.162	0.928–1.455	0.191
Tumor rupture Yes vs. No	1.914	1.338–2.737	<0.001 ***	1.326	0.872–2.017	0.187
Cirrhosis Yes vs. No	1.376	1.123–1.686	<0.001 ***	1.417	1.133–1.771	0.002 **
AJCC 8th Stage III vs. II vs. I	1.809	1.590–2.058	<0.001 ***	NA	NA	NA

* statistical significance (* *p* < 0.05, ** *p* < 0.01, *** *p* < 0.001); HR, hazard ratio; 95% CI, 95% confidence interval of hazard ratio. Disease-free survival was calculated by univariate and multivariate Cox regression analysis; AST: aspartate aminotransferase; ALT: alanine aminotransferase; AFP: alpha-fetoprotein; AJCC 8 staging: the 8th edition of American Joint Committee on Cancer TNM staging system.

**Table 4 biomedicines-12-00655-t004:** Clinicopathologic data of 754 patients with HCC in univariate and multivariate regression analysis in relation to OS.

Variables	Univariate Analysis			Multivariate Analysis		
	HR	95% CI	*p* Value	HR	95% CI	*p* Value
Age (60 years) >60 vs. ≤60	1.101	0.826–1.462	0.507			
Sex (M/F) M vs. F	1.393	0.962–2.016	0.079			
Diabetes Yes vs. No	1.568	1.159–2.121	0.004 **	1.657	1.214–2.263	0.001 **
Comorbidity Yes vs. No	1.276	0.953–1.707	0.102			
ICG-R15 (10%) Higher vs. lower	1.060	0.737–1.524	0.754			
AST (68 U/L) Higher vs. lower	1.934	1.351–2.768	<0.001 ***	1.249	0.847–1.842	0.263
ALT (72 U/L) Higher vs. lower	1.319	0.884–1.970	0.175			
AFP (1000 ng/mL) Higher vs. lower	1.755	1.201–2.565	0.004 **	1.142	0.760–1.716	0.522
Major hepatectomy Yes vs. No	1.893	1.418–2.525	<0.001 ***	1.075	0.762–1.517	0.678
Anatomic resection Yes vs. No	1.072	0.807–1.424	0.630			
Close margin Yes vs. No	1.280	0.962–1.706	0.091			
Complication Yes vs. No	2.617	1.692–4.048	<0.001 ***	2.086	1.307–3.329	0.002 **
Tumor size (cm) >5.0 vs. ≤5.0	2.758	2.076–3.665	<0.001 ***	1.271	0.865–1.866	0.222
Satellite lesions Multiple vs. single vs. no	1.766	1.478–2.236	<0.001 ***	0.997	0.757–1.313	0.982
Vascular invasion Thrombus vs. microscopic vs. no	1.818	1.478–2.236	<0.001 ***	0.913	0.669–1.245	0.565
Grading I/ II/ III, IV III, IV vs. I, II	1.411	1.064–1.872	0.017 *	1.138	0.840–1.543	0.404
Tumor rupture Yes vs. No	2.322	1.502–3.589	<0.001	0.646	0.378–1.103	0.109
Cirrhosis Yes vs. No	1.478	1.110–1.968	0.007 **	1.223	0.900–1.331	0.198
AJCC 8th Stage III vs. II vs. I	2.008	1.685–2.393	<0.001 ***	1.474	1.086–2.001	0.013 **
Recurrence type III vs. II vs. I	2.144	1.919–2.395	<0.001 ***	1.955	1.733–2.205	<0.001 ***

* statistical significance (* *p* < 0.05, ** *p* < 0.01, *** *p* < 0.001); HR, hazard ratio; 95% CI, 95% confidence interval of hazard ratio. Disease-free survival was calculated by univariate and multivariate Cox regression analysis; AST: aspartate aminotransferase; ALT: alanine aminotransferase; AFP: alpha-fetoprotein; AJCC 8 staging: the 8th edition of American Joint Committee on Cancer TNM staging system.

**Table 5 biomedicines-12-00655-t005:** The impact of histology on recurrence pattern.

	No Recurrence	Solitary Intrahepatic Lesion	Multiple Intrahepatic Metastasis	Any Distant Metastasis
*p* = 0.002				
G1 + G2	54.4%	30.9%	7.2%	7.4%
G3 + G4	44.8%	33.8%	6.2%	15.3%
*p* = 0.002				
G1	56.2%	30.6%	9.9%	3.3%
G2	53.7%	31.1%	6.2%	9.0%
G3	46.6%	32.7%	6.4%	14.2%
G4	25.9%	44.4%	3.7%	25.9%

G: Edmondson–Steiner grade.

## Data Availability

The datasets used and analyzed during the current study are available from the corresponding author on reasonable request.

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
