# Peer review of "Recurrence Pattern Is an Independent Surgical Prognostic Factor for Long-Term Oncological Outcomes in Patients with Hepatocellular Carcinoma"

_biomedicines, 2024, doi:10.3390/biomedicines12030655_

Round 1

Reviewer 1 Report

Comments and Suggestions for Authors

This study examined the recurrence patterns following curative hepatectomy for hepatocellular carcinoma. A retrospective analysis of 754 patients revealed three distinct recurrence groups: regional recurrence, multiple intrahepatic recurrence, and distant metastasis. Results showed that distant metastasis recurrence was more common in males and those with specific tumor characteristics. Patients with distant metastasis had the shortest recurrence time and the poorest overall survival compared to other recurrence patterns. This suggests that distant metastasis recurrence poses the greatest risk and has the worst outcomes among HCC patients following curative hepatectomy. Some revisions are suggested.

1. It would be beneficial if the authors could present pathological evidence during the time of HCC resection. Are there any pathological features that can predict the recurrence pattern of HCC?

2. Are the primary diseases and the severity of fibrosis associated with recurrence?

3. Do different surgical treatment methods have an impact on the likelihood of recurrence occurring?

4. It would be beneficial if the author could offer treatment strategies tailored to patients with different recurrence patterns. This could involve outlining specific therapeutic approaches based on the distinct types of recurrence observed in HCC patients.

5. A risk factor analysis between recurrence and non-recurrence patients would be appreciated.

Comments on the Quality of English Language

Minor editing.

Reviewer 2 Report

Comments and Suggestions for Authors

The manuscript entitled " Recurrence pattern is an independent surgical prognostic factor for long-term oncological outcomes in patients with hepatocellular carcinoma " has been reviewed.

This paper shows that the pattern of recurrence of hepatocellular carcinoma is an independent surgical prognostic factor for long-term oncological outcome.

Although the content is not new, it is an interesting paper.

Do the authors recommend neoadjuvant or adjuvant chemotherapy or even both in cases of men, extended liver resection and vascular invasion, etc., as shown in their analysis of type III recurrent patients?

Does the histological type of cancer have any influence on metastasis?

Comments on the Quality of English Language

none

Round 2

Reviewer 1 Report

Comments and Suggestions for Authors

My questions have been well-addressed.